AI; dementia; deep learning; machine learning; cognition

**Author for correspondence:**
Kelvin K. F. Tsoi,
Email: kelvintsoi@cuhk.edu.hk

# Applications of artificial intelligence in dementia research

Kelvin K. F. Tsoi[1,2], Pingping Jia[1], N. Maritza Dowling[3,4], Jodi R. Titiner[5], Maude Wagner[6], Ana W. Capuano[6] and Michael C. Donohue[7]

[1]JC School of Public Health and Primary Care, The Chinese University of Hong Kong, Sha Tin, Hong Kong; [2]Stanley Ho Big Data Decision Analytics Research Centre, The Chinese University of Hong Kong, Sha Tin, Hong Kong; [3]Department of Acute and Chronic tableCare, School of Nursing, The George Washington University, Washington, DC, USA; [4]Department of Epidemiology and Biostatistics, Milken Institute School of Public Health, The George Washington University, Washington, DC, USA; [5]Alzheimer's Association, Chicago, USA; [6]Department of Neurological Sciences, Rush Alzheimer's Disease Center, Rush University Medical Center, Chicago, IL, USA and [7]Alzheimer's Therapeutic Research Institute (ATRI), University of Southern California, Los Angeles, CA, USA

## Abstract

More than 50 million older people worldwide are suffering from dementia, and this number is estimated to increase to 150 million by 2050. Greater caregiver burdens and financial impacts on the healthcare system are expected as we wait for an effective treatment for dementia. Researchers are constantly exploring new therapies and screening approaches for the early detection of dementia. Artificial intelligence (AI) is widely applied in dementia research, including machine learning and deep learning methods for dementia diagnosis and progression detection. Computerized apps are also convenient tools for patients and caregivers to monitor cognitive function changes. Furthermore, social robots can potentially provide daily life support or guidance for the elderly who live alone. This review aims to provide an overview of AI applications in dementia research. We divided the applications into three categories according to different stages of cognitive impairment: (1) cognitive screening and training, (2) diagnosis and prognosis for dementia, and (3) dementia care and interventions. There are numerous studies on AI applications for dementia research. However, one challenge that remains is comparing the effectiveness of different AI methods in real clinical settings.

## Impact statement

Artificial intelligence becomes popular for dementia research. Supervised and unsupervised machine learning models can be applied for cognitive screening, diagnosis, prognosis, and potentially for dementia care and treatment development.

## Introduction

*Dementia* is a syndrome characterized by deterioration of cognitive function and behavior beyond what might be expected from the usual consequences of biological aging (Ernst and Hay, 1994; Bouchard, 2007). The prevalence rate among those aged ≥60 years in different world regions is approximately 5%–7% (Prince et al., 2013), and the total number of dementia cases is expected to increase from 57.4 million in 2019 to approximately 150 million by 2050 (Nichols et al., 2022). The etiological subtypes of dementia include Alzheimer's disease (AD), vascular dementia, frontotemporal dementia (FTD), frontotemporal lobar dementia, Huntington's disease, Lewy bodies, and Parkinson's disease (Bouchard, 2007). Mild cognitive impairment (MCI) is also regarded as an early stage of dementia (Petersen, 2004; Morris, 2006). AD accounts for approximately 60%–70% of diagnosed dementia cases and attracts the most attention from researchers. Early detection and timely diagnosis are challenging as the diagnoses of dementia are based on a comprehensive procedure of semi-structured interviews, cognitive tests, and medical examinations, which are time-consuming, costly, and sometimes even invasive. AD occurs mainly in the elderly, and it is difficult to distinguish between a degenerative condition and the general impact of aging. Approximately 29%–76% of individuals with dementia are unrecognized in clinical practice (Valcour et al., 2000; Knopman et al., 2003; Chodosh et al., 2004).

Researchers employed artificial intelligence (AI) in clinical decision support systems, new therapy discovery, and genomics research by using different biomarkers of dementia (Miao et al., 2017; Zhu et al., 2020; Anastasio, 2021). Those biomarkers are measurable indicators of a biological state for dementia or cognitive decline, including neuroimaging, retinal imaging, language information, cerebrospinal and blood biomarkers, and gene information. AI has led to great breakthroughs in image processing (Krizhevsky et al., 2012) and natural language

processing (NLP), such as the development of speech-to-speech translation engines and spoken dialogue systems (Hirschberg and Manning, 2006) that allow for more complicated and fast analysis of neuroimaging and speech data. For complex data sources such as magnetic resonance imaging (MRI), position emission tomography (PET) neuroimaging, and cerebrospinal fluid (CSF) biomarkers (Suk et al., 2016; Ebrahimighahnavieh et al., 2020), deep learning methods (e.g., neural network-related methods) can be applied to build diagnostic classifiers or applied in feature extraction steps (e.g., Auto-Encoder; Ebrahimighahnavieh et al., 2020). The support vector machine (SVM) algorithm is the most widely used machine learning method to classify diseases like Alzheimer's, Epilepsy, and Parkinson's (Deepa et al., 2017). The neural network-based methods are also popular, including multilayer perception and deep learning methods such as the convolutional neural network (CNN). An upsurge in chemical data availability makes AI viable for virtual screening (VS) of drug discovery (Vamathevan et al., 2019). The global popularity of smartphones makes it possible to deploy a variety of mobile apps for cognitive training and screening, including traditional cognitive tests and unconventional new methods (Thabtah et al., 2020; Chelberg et al., 2021). Social robots were developed to assist dementia patients in performing basic instrumental activities of daily living (Schroeter et al., 2013; Law et al., 2019; Ghafurian et al., 2021).

This review summarizes AI in dementia research and its application to (i) dementia diagnosis and prognosis, (ii) cognitive screening and training, and (iii) care and treatment. The paper search was conducted on Ovid Embase, MEDLINE, Web of Science, IEEE Xplore, and ScienceDirect with the following keywords: artificial intelligence or machine learning or deep learning or support vector machine or decision tree and dementia or cognitive or Alzheimer. Papers that applied machine learning technology on dementia were included whether they were original study or reviews.

## Basic concepts of artificial intelligence

*AI* is a general term that means imitating intelligent human behavior using a computer with minimal human intervention. Research into AI applications began shortly after the official naming of AI at a Dartmouth College meeting in 1956 (Mishra and Li, 2020). Machine learning is a subfield of AI that works by examining and learning patterns of input datasets to build models for classification, regression, and clustering. Deep learning, reinforcement learning, and transfer learning are more specific subsets of machine learning. As reported in Table 1, machine learning methods include SVM, random forest (RF), *k*-nearest neighbor (*k*-NN), and so on. Deep learning methods are neural network-based methods, such as CNN and artificial neural network (ANN). It has only recently become a trend with the onset of the "Big Data" era, although it has existed since the 1950s.

Machine learning can be divided into three types: supervised, unsupervised, and semi-supervised learning, according to whether the input data are labeled (Kumar et al., 2021). Supervised learning means training a model on a dataset annotated with labels applied in classification and regression tasks, such as linear regression (LR) and SVM. In comparison, unsupervised models learn from unlabeled data by extracting features and patterns in solving clustering problems, such as *k*-NN and principal component analysis. Finally, the semi-supervised method builds a model on a training dataset with labels in one part and no labels in the other. A specific

algorithm does not necessarily belong to only one of the three types. For example, semi-supervised SVM is a good solution when the datasets contain unlabeled data, whereas the standard SVM cannot perform well in this situation (Ding et al., 2017).

## Applications in dementia diagnosis and prognosis

The complex process of dementia diagnosis involves a combination of medical examinations and professional clinicians. The Diagnostic and Statistical Manual of Mental Disorders (DSM) is widely used as a general diagnostic criterion for dementia, and the most updated version is DSM-5 (Bouchard, 2007; American Psychiatric Association, 2013). However, it does not represent all the clinical profiles of some subtypes of dementia, such as vascular and FTD, where memory impairment is not necessarily the first requirement. Currently, there are also other different diagnostic criteria which are used in clinical settings, and dementia research (e.g., NINCDS-ADRDA10 criteria and 10th revision of the International Classification of Diseases; McKhann et al., 1984; World Health Organization, 1992; Bouchard, 2007). Cognitive deficits might appear in many other diseases, but only those diseases whose core features are cognitive disorders and decline are included as neurocognitive disorders. Diagnosis involves cognitive function, language, praxis, gnosis, executive function, and other medical tests. Overall, there are no perfect criteria, and the diagnostic process is complicated. A large number of patients remain undiagnosed due to costly and time-consuming procedures (Prince et al., 2016). The disease progresses, before symptoms appear clearly, while patients experience mild-to-moderate cognitive impairment.

Machine learning methods are widely applied in high-dimensional clinical data for dementia prediction (Spooner et al., 2020). The Alzheimer's Disease Prediction of Longitudinal Evolution Challenge was held in 2017, which aims to identify the most effective features and approaches that predict clinical diagnosis of AD, Alzheimer's Disease Assessment Scale Cognitive Subdomain (ADAS-Cog13), and total volume of the ventricles (Marinescu et al., 2021). The challenge compared the performance of 92 algorithms and found that those algorithms perform differently on three outcomes, and CSF samples and diffusion tensor imaging were associated with better diagnosis performance (Marinescu et al., 2021). With the number of publications increasing drastically since 2017 (Ebrahimighahnavieh et al., 2020), deep learning has begun to gain considerable attention in research for AD detection. The data sources applied in AI models are beyond the traditional data format, such as age, gender, and comorbidity. New forms of cognitive data include neuroimaging, speech and language, genetic research, CSF and blood biomarkers, and electroencephalogram (EEG) and retinal imaging (Figure 1).

### *Neuroimaging*

Specific changes in brain structure and other metabolite responses in the brain can be measured by modern techniques, such as PET and MRI. Machine learning methods have been developed to detect disease via neuroimaging with improved medical imaging and greater availability of neuroimaging data (Pellegrini et al., 2018). Due to the complexity of imaging data, feature extraction for neuroimaging data is still a challenge for data scientists. Feature extraction of imaging data can be generally grouped into four categories: voxel-based, slice-based, patch-based, and regions-of-interest (ROIs)-based features (Ebrahimighahnavieh et al., 2020).

**Table 1.** Abbreviation for machine learning methods*

| Abbreviation | Methods |
| --- | --- |
| SVM | |
| SVM | Support vector machine |
| RBF-SVM | Radial basis function |
| Discriminant analysis | |
| LDA | Linear discriminant analysis |
| QDA | Quadratic discriminant analysis algorithm |
| Tree model | |
| DT | Decision tree |
| RF | Random forest |
| AdaBoost | Adaptive boosting |
| XGBoost | Extreme gradient boosting |
| Bayes model | |
| NB | Naïve Bayes |
| GNB | Gaussian naïve Bayes |
| Regression | |
| LR | Logistic regression |
| MLR | Multiple linear regression |
| LASSO | Least absolute shrinkage and selection operator |
| PLSR | Polynomial least squares regression |
| Neural network-based | |
| MLP | Multilayer perception |
| RBM | Restricted Boltzmann machine |
| AE | Autoencoder |
| ANN | Artificial neural network |
| DNN | Deep neural network |
| DPN | Deep polynomial network |
| CNN | Convolutional neural network |
| RNN | Recurrent neural network |
| RBF-NN | Radial basis function network |
| DBN | Deep belief network |
| Other | |
| SRC | Sparse representation classification |
| GC | Gaussian classifier |
| k-NN | k-nearest neighbor |
| Factor analysis | Factor analysis |
| OPLS | Orthogonal projections to latent structures |
| – | Hierarchical clustering |
| – | Bayesian network |
| – | Ensemble neural network |
| ICA | Independent component analysis |

*The list only shows machine learning methods reported in this study.

ROI-based and patch-based methods were reported to be better pre-processing methods as they can exclusively include AD-related features in neuroimaging (Ebrahimighahnavieh et al., 2020).

The advanced processing power of Graphics Processing Units makes it possible to apply deep learning methods in neuroimaging, particularly for CNNs, showing good performance in detecting disease by medical imaging (Ebrahimighahnavieh et al., 2020). Pellegrini et al. (2018) reviewed studies from 2006 to 2016 and eventually included 111 studies of machine learning of neuroimaging on dementia detection. More than half of those studies applied SVM, and other methods, such as adaptive boosting (AdaBoost), linear discriminant analysis, and RF, were also involved. In the recent decade, deep learning has been a trend in the neuroimaging data processing. It imitates the working of the human brain and can merge complicated feature extraction and classification in solving complex problems. A review focused on deep learning and neuroimaging included more than 100 papers, all of which were conducted after 2013 and most (80%) after 2017 (Ebrahimighahnavieh et al., 2020). Hidden Markov Model, one of the reinforcement learning methods, is especially suitable for detecting the progression of dementia by analyzing sequence neuroimaging data, and it has been applied by different research groups (Chen and Pham, 2013; Williams et al., 2019). Most studies pay closer attention to technical details, but have less of a clinical focus (Pellegrini et al., 2018). A clinical-based framework is, therefore, needed to improve the application of the AI models.

## Speech and language

Recent studies have suggested that language dysfunction is one of the earliest signs of cognitive disorders and a possible biomarker for the early detection of dementia (Ahmed et al., 2013; Mueller et al., 2017; Beltrami et al., 2018; Garcia et al., 2020). Speech and language have long been used as important clinical information for dementia diagnosis, such as the Boston naming test studied and reported since 1986 (Knesevich et al., 1986). It can be obtained through content-based (specific tasks) (Rodriguez-Aranda et al., 2016; Venneri et al., 2018) or content-free approaches (spontaneous conversation) (Huff, 1986; Becker et al., 1994). The dysfunction includes word retrieval difficulties (e.g., verbal naming, accurate meaning communication, and pulsation) and a tendency to repeat words or sentences. NLP plays an essential role in speech and text data analysis to extract prosodic, acoustic, or other features in dementia analysis (Jaffe and Feldstein, 1970; Forbes et al., 2002; Weiner et al., 2017; Luz et al., 2018). Researchers widely transformed NLP in the 1990s to build models over large quantities of empirical language data (Hirschberg and Manning, 2006). Traditional text feature extraction includes filtration, fusion, mapping, and clustering, which result in a lengthy process. Deep learning can be used to quickly acquire effective characteristics from training data, and CNN and recurrent neural networks (RNN) are two popular models (Liang et al., 2017). Early NLP on the text-based dialogue has expanded to include spoken dialogue with the development of spoken dialogue systems and automatic speech recognition, which allows a more effective speech feature extraction (Weiner et al., 2017).

Although many studies were conducted in building diagnostic models with a series of machine learning methods, such as SVM, decision tree, and RF, the expansion of those models is limited due to the small sample size and incomparable datasets. In 2020, the Alzheimer's Dementia Recognition through Spontaneous Speech Challenge at INTERSPEECH 2020 provided an opportunity to use all available audio and textual data from a benchmark speech dataset. The challenge defined shared tasks and provided a standardized dataset based on spontaneous speech and allowed different

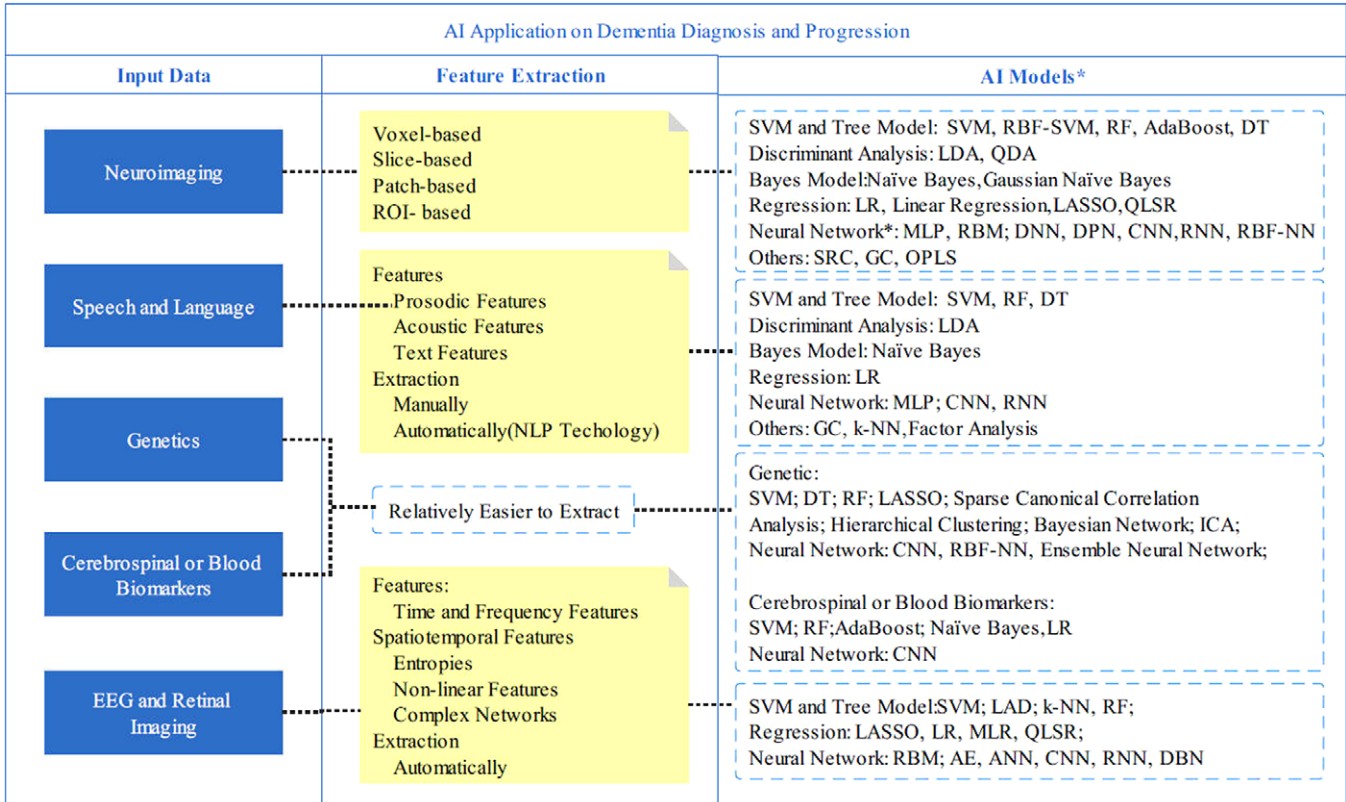

**Figure 1.** Applications of AI on Dementia Diagnosis and Prognosis

research groups to test the performance of their existing or novel methods (Luz et al., 2020).

### Genetics

Approximately 58%–79% of the risk of late-onset Alzheimer's is heritable (Gatz et al., 2006; Sims et al., 2020). Studies suggest that patients with FTD (Greaves and Rohrer, 2019) or Lewy bodies dementia (Guerreiro et al., 2019) have a high proportion of positive family history in as many as 60% of cases (Mackenzie et al., 2014). High-dimensional genetic-related data have been generated with the evolution of genomics. AI methods, such as SVM, have shown good performance on gene identification and pathway analysis, and RF is well suited for microarray data (Miao et al., 2017). Xu et al. (2018) applied the SVM model to predict AD by using protein sequence, and the accuracy rate was 85.7%, where RF, naïve Bayes, AdaBoost, and Bayes network were also applied and compared with the result of SVM. Varatharajah et al. (2019) integrated genetic data, multimodal neuroimaging, CSF biomarkers, genetic factors, and measures of cognitive resilience data to build an SVM model to predict the progress of MCI to AD within 3 years with an accuracy rate up to 93%. Machine learning methods can also be applied in detecting significant genetic variants, gene expression, and gene–gene interaction (Mishra and Li, 2020).

### Cerebrospinal fluid and blood biomarkers

Measuring Amyloid-Beta, total tau (T-tau), and hyperphosphorylated tau (p-tau) in CSF proteins has proved accurate in diagnosing AD (Ritchie et al., 2017; Zetterberg, 2019). However, these biomarkers are expensive and relatively invasive. In addition to invasive methods, imaging technology can reflect the level of CSF biomarkers (e.g., tau PET). With the recent advent of highly sensitive and specific immune and mass spectrometry-based assays, CSF biomarkers can also be detected through blood (Varma et al., 2018). Other blood biomarkers include *N*-methyl-D-aspartate receptor-mediated biomarkers and metabolites biomarkers. The first d-glutamate-based study that applied machine learning models to detect MCI and AD in healthy people was published in 2021 (Chang et al., 2021). A total of 133 AD patients, 21 MCI patients, and 31 healthy controls were recruited, and four machine learning algorithms (SVM, LR, RF, and Naïve Bayes) were employed to build predictive models to distinguish MCI or AD patients from healthy controls, with sex, age, and d-glutamate as predictors. The Naïve model and the RF model showed the best performance with area under the curve (AUC) of 0.82 and 0.79 (Chang et al., 2021). Stamate et al. (2019) applied deep learning, extreme gradient boosting, and RF on plasma metabolites data to differentiate healthy people and patients with AD. They also showed better AUC than the results from amyloid, p-tau, and t-tau. Although the AUC seems good, which is usually more than 0.80, most studies lack external validation. Further studies are needed to assess the performance of combinations of CSF, blood biomarkers, and lifestyle factors.

### Electroencephalogram and retinal imaging

An EEG is a test, administered by hospital equipment and wearable devices, that detects abnormalities in brain waves or in the electrical activity of the brain. In the recent decade, researchers have set their sights on AD diagnosis and progression based on EEG data ((Malek et al., 2017; Stancin et al., 2021). Due to the complexity of EEG data,

feature extraction is a crucial step, including time-domain and frequency-domain features, nonlinear features, entropies, spatio-temporal features, and complex networks (Stancin et al., 2021). Each feature contains a series of the index and is explored by many studies (Deepa et al., 2017; Jaya Shree and Venkateshwarlu, 2021). SVM is widely used for binary classification (Staudinger and Polikar, 2011; Jaya Shree and Venkateshwarlu, 2021; Tzimourta et al., 2021). Sharma et al. (2021) conducted a multiclass SVM in 2021 with a diagnostic accuracy of 87.6%, in which they initially extracted 12 EEG features and then selected five of them through analysis of variance. Deep learning methods, such as RNN and ANN, are rapidly increasing in EEG studies. Ieracitano et al. (2020) proposed a multimodal machine learning that integrated Multilayer Perceptron, LR, and SVM to classify MCI and dementia using EEG data.

Retinal imaging is a cost-effective replacement for neuroimaging as retinal changes can reflect the pathology of the brain. The quantitative analysis of vessel calibers, tortuosity, and network complexity in retinal imaging data provide diagnostic value for dementia. Tian et al. (2021) proposed a multistage pipeline that involved SVM and CNN and achieved an average diagnostic accuracy of 82.4% for AD.

## Applications in cognitive screening and training

Several mobile apps have been developed to screen normal individuals' cognitive function before they suffer from MCI or dementia (Thabtah et al., 2020). Most mobile apps assist individuals diagnosed with MCI or dementia in brain training. In addition to apps, machine learning can also contribute to building new assessments for MCI or dementia. Chiu et al. (2019) applied information gain, which is a feature selection method in RF, in developing a brief questionnaire to help clinicians in dementia diagnosis. With advances in wearable technologies, plenty of data collected from wearable sensors can also be applied in machine learning models and thus improve the performance (Iaboni et al., 2022).

### Computerized cognitive screening

Dementia screening aims to identify those in the prodromal phase of dementia by using neuropsychological tests (Panegyres et al., 2016). The tests include the Abbreviated Mental Test, the Montreal Cognitive Assessment (MoCA), the Mini-Mental State Examination (MMSE), and others. Several dementia screening methods are now available on mobile apps, making them more accessible to patients, caregivers, and medical staff. Those screening tools could be divided into three categories (Thabtah et al., 2020): (1) apps based on a single medical assessment method (such as MMSE and MoCA), (2) apps based on multiple medical assessment methods (e.g., DementiaTest, which integrated six-item cognitive impairment and the structured clinical interview; Thabtah et al., 2019), and (3) apps based on nonconventional methods. Cognity is one of the eligible apps, which applied AI technology to screen for AD by combining analysis of a clock photo drawing by the user and the Mental Status Examination (Thabtah et al., 2020).

A systematic review in 2020 evaluated mobile apps of dementia screening available on Android and Apple platforms (Thabtah et al., 2020). The evaluated criteria were based on DSM-5, including six domains of cognitive function. They initially found 275 apps in the English language, and only 20 apps were eligible. Most excluded apps were games and informative apps to assist individuals in their cognitive functions and skills (Thabtah et al., 2020). Another systematic review performed by Chan et al. (2021) evaluated the diagnostic performance of digital tests, finding a few validation studies for all digital tests, and the eligible apps had a sensitivity and specificity of more than 0.8.

### Computerized cognitive training

Cognitive training via digital devices is a promising strategy for maintaining the cognitive function of healthy elderly and MCI patients (Zhang et al., 2019). The main advantages are the active accessibility and timely feedback (Irazoki et al., 2020). In addition to providing cognitive training for normal people or MCI patients, most training apps focus on caregivers or family members of dementia patients to assist them in caring for dementia patients. A review performed by Chelberg et al. (2021) included 75 Australian-based apps, focusing on cognitive training and addressing care needs. The majority of them were free to download, and their primary audience were caregivers, with approximately 40% of them focusing on MCI or dementia patients (Chelberg et al., 2021).

### Others

In addition to those digital games, serious games were designed and developed for cognitive screening. Users were motivated and engaged to regularly perform screening tasks by playing serious games (Cha et al., 2019). Karapapas and Goumopoulos (2021) applied machine learning methods using demographic characteristics and data collected from serious games, which showed a high detection performance. The flexibility of wearable platforms has also provided a variety of data to detect cognitive status (Iaboni et al., 2022).

## Applications in dementia care and treatment

### Socially assistive robots

Given the complexity of dementia care and that the aging population requires more care from a decreasing number of caregivers, researchers have been exploring ways to utilize advanced robotic technology to assist elderly care (Hung et al., 2019; Koutentakis et al., 2020). PARO is one of the most popular interactive pet robots for older adults. It has the appearance of a baby harp seal and provides companionship and emotional interaction to users (Hung et al., 2019). Other socially interactive robots provide support in daily engagement for those who have MCI or are at the early stage of dementia (Law et al., 2019). CompanionAble is a robot that helps MCI or dementia patients live at home by linking to a smart home environment (Schroeter et al., 2013). This robot focuses on cognitive and social support, such as daily activity reminders, suggesting activities, video calling, and cognitive training, which were tested with five couples in their homes over 2 days and potentially reduced the burden for caregivers (Schroeter et al., 2013). RobuLAB10 is a robot to monitor emotions, help in health emergencies, make calls, and provide cognitive training and other support for daily activities (Pino et al., 2015). These social robots are tested in research with limited participants followed during a short period and are not widely adopted in the real world.

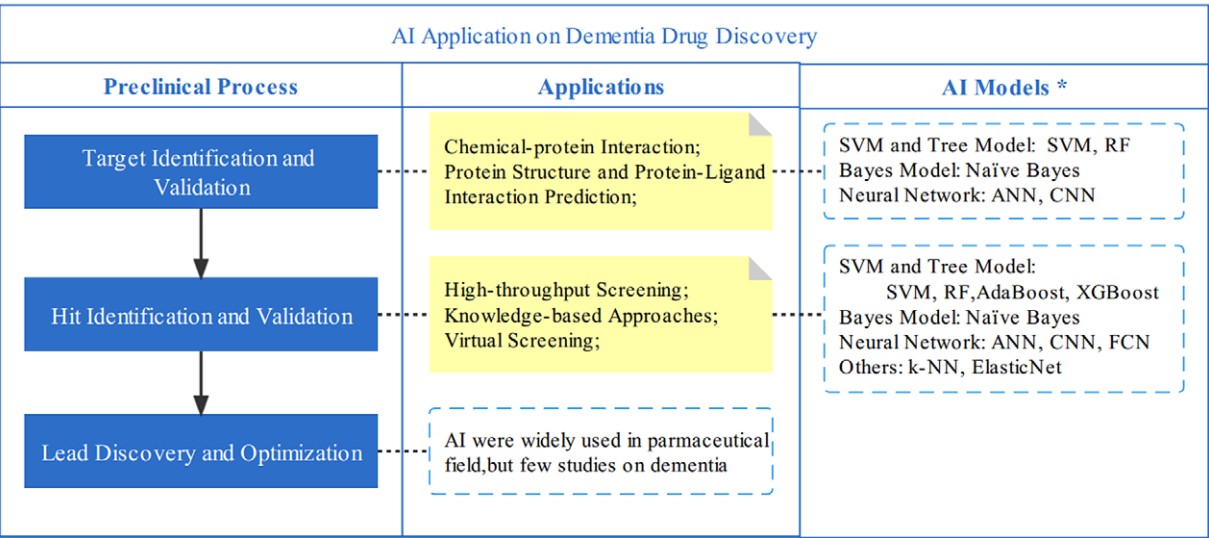

**Figure 2.** Applications of AI on Dementia Drug Discovery

### *Drug discovery*

New drug discovery is a long process, including preclinical processes and clinical trials. The preclinical process before in vitro tests includes target identification and validation, compound screening, and lead discovery (Stamate et al., 2019). AI techniques could be applied in all aspects of this process to accelerate new drug development, among which VS and target discovery are the most common application scenarios (Stamate et al., 2019). Figure 2 shows the common AI application of dementia drug discovery.

Target discovery aims to confirm a causal association between target and disease, which involves protein structure and chemical–protein interaction predictions (Fang et al., 2015). Recent studies indicated that AD shares intermediate endophenotypes and underlying mechanisms with other diseases, which means there would be multiple targets for AD (Fang et al., 2020). Fang et al. applied naïve Bayesian and recursive partitioning algorithms to predict active compounds bound to as many targets as possible (e.g., the Amyloid Precursor Protein). The methods were evaluated with internally fivefold cross-validation and an external test dataset with an average AUC of 0.965 (Fang et al., 2015). His team also proposed a network-based AI framework to identify potential drug targets by integrating multiomics data, human protein–protein networks, and other related data (Fang et al., 2021).

VS is extremely computationally intensive and likely to take an incredible amount of time in silico searches over millions of compounds, ultimately increasing yields of potential drug lead (Carpenter and Huang, 2018). Machine learning methods are conducted to speed this process up by building predictive models using active and inactive molecules. SVM is generally among the top performers in machine learning for VS studies (Carpenter and Huang, 2018). It applied the "kernel" function to map the database molecules into high-dimensional representations. Yang et al. (2010) performed SVM and RF to predict γ-secretase inhibitors and noninhibitors related to AD prevention and treatment.

Deep learning is reported to perform better than machine learning methods (Carpenter and Huang, 2018), among which ANN and CNN are the most widely used methods (Anastasio, 2021; Fang et al., 2022; Wang et al., 2022).Rodriguez et al. (2021) proposed a machine learning framework to nominate drugs that the FDA had already approved. They explored the potential associations between AD and molecular mechanisms described by a list of genes, which integrated LR, SVM, RF, and two-layer CNN. Wang et al. (2022) applied graph-based deep learning for drug–target interaction prediction, which performed better than naïve Bayes, logistic regression, and RF classifiers.

### Conclusions and future challenges

AI technology can be applied to the field of dementia research to contribute to fast and accurate diagnosis, providing accessible cognition training tools and reducing care burden. AI can also monitor the progression from MCI to dementia so that those at high risk receive timely interventions. Deep learning has become more popular, particularly in image processing, due to its ability to process complex data. However, one the one hand, the complexity algorithms and black box of explanation restrict its access to clinical researchers. On the other hand, most of the studies explored the performance of the proposed algorithms on datasets with different sample sizes and data features, which makes it challenging to compare different methods. Online datasets, such as ADNI, AIBL, and MIRIAD, have contributed to dementia research (Ebrahimighahnavieh et al., 2020). However, even in the same datasets, different studies may still be incomparable as they can apply different parts of those data for different features. For example, applying single biomarkers or merging various data sources, such as demographic and clinical information, performs differently. Thus, further exploration of benchmarking datasets and standard frameworks in different types of dementia is a necessary challenge. In addition, the increasing usage of smart wearable devices that generate complexity big data has provided a new avenue for detecting cognitive impairment (Chen et al., 2019). This kind of digital biomarker has enormous potential power in dementia research in the future.

**Open peer review.** To view the open peer review materials for this article, please visit http://doi.org/10.1017/pcm.2022.10.

**Acknowledgements.** This manuscript was facilitated by the Alzheimer's Association International Society to Advance Alzheimer's Research and

Treatment (ISTAART), through the Design and Data Analytics professional interest area (DaDA PIA). The views and opinions expressed by the authors in this publication represent those of the authors and do not necessarily reflect those of the PIA membership, ISTAART, or the Alzheimer's Association.

**Author contributions.** K.K.F.T. and P.J. authors made an equal contribution to this work.

**Competing interest.** The authors declare none.

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
