## [Reviewer Report]

*Comments to Author*: This paper provides an overview of several application of AI in dementia. The authors present recent work in three different areas: (i) dementia diagnosis and prognosis; (ii) cognitive screening and training and (iii) dementia care and treatment.

Most of the referenced works are very recent in time, which demonstrates that this is a very active research field.

Nevertheless, this paper lacks the required academic rigor for research reviews and surveys. As it is written, it is not clear what the criteria to select works presented was. For example, there important screening tools that combina AI with serious games or conversational agents that are not mentioned at all even they are recent publication at top quality journals. To avoid this kind of issues, analysis of state-of-the-art is performed through systematic reviews that query the most relevant databases and are reproducible by others, so that, there is no subjectivity as far selection of works is concerned.

As it is, my opinion is that this work cannot be published. It is an overview of the application of AI in three specific dementia-related domains. But it does not provide any additional value, neither for those from the clinical side nor for those on the computational side. The conclusions are not adequately concrete and are not sufficiently based on the data raised from the works review.

My suggestion is carry out systematic reviews - for instance following the PRISMA methodology - focusing on one or ore of the domains identified. A meta-review is also an option but, again, this requiere a systematic approach when querying databases to identify works to be included in the review.

---

## [Reviewer Report]

*Comments to Author*: This is a review paper intended to provide an overview of AI applications in dementia research. Overall, this review is well written and generally reasonable in terms of categorizing AI applications in dementia research. However, in order to improve the quality of the review article, the following points should be modified.

Major:

1) Since this is a review article, the authors must explain in detail what criteria they used to evaluate the study for inclusion in the review, i.e., what selection criteria they used and from what data sources they took the article. In recent years, research articles on medical AI have been published not only in medical journals, but also in informatics journals and proceedings, and are not necessarily indexed in Pubmed.

2) Several recent studies have proposed methods to monitor symptoms related to cognitive impairment by integrating data from multiple consumer-grade smart devices (e.g., Chen, et. al., 2019, doi: https://doi.org/10.1145/ 3292500.3330690.)

The authors should consider to include some description of such screening studies using routine measuring devices.

---

## [Editor Report]

*Comments to Author*: The two reviewers have made very similar comments on the manuscript. The authors need to make major revisions accordingly.

---

## [Editor Report]

*Comments to Author*: The authors have made substantial revisions based on the reviewer's comments. Reviewer #2 was satisfied with the current draft. The paper is ready to be published.